# AN APPLICATION OF PSEUDO-LOG-LIKELIHOODS TO NATURAL LANGUAGE SCORING

## ABSTRACT

Language models built using semi-supervised machine learning on large corpora of natural language have very quickly enveloped the fields of natural language generation and understanding. In this paper we apply a zero-shot approach independently developed by a number of researchers now gaining recognition as a significant alternative to fine-tuning for evaluation on common sense tasks. A language model with relatively few parameters and training steps (albert-xxlarge-v2) compared to a more recent language model (T5) can outperform it on a recent large data set (TimeDial), while displaying robustness in its performance across a similar class of language tasks. Surprisingly, this result is achieved by using a hyperparameter-free zero-shot method with the smaller model, compared to fine-tuning to the larger model. We argue that robustness of the smaller model ought to be understood in terms of compositionality, in a sense that we draw from recent literature on a class of similar models. We identify a practical cost for our method and model: high GPU-time for natural language evaluation. The zero-shot measurement technique that produces remarkable stability, both for ALBERT and other BERT variants, is an application of pseudo-log-likelihoods to masked language models for the relative measurement of probability for substitution alternatives in forced choice language tasks such as the Winograd Schema Challenge, Winogrande, CommonsenseQA, and others. One contribution of this paper is to bring together a number of similar, but independent strands of research. We produce some absolute state-of-the-art (SOTA) results for common sense reasoning in binary choice tasks, performing better than any published result in the literature, including fine-tuned efforts. In others our results are SOTA relative to published methods similar to our own – in some cases by wide margins, but below SOTA absolute for fine-tuned alternatives. In addition we show a remarkable consistency of the model's performance under adversarial settings, which we argue is best explained by the model's compositionality of representations.

## 1 INTRODUCTION

Computational linguistics has made major strides in the adoption of machine learning techniques applied to unstructured corpora consisting of human generated natural language text. For example, some methods take advantage of the frequencies of words in natural human text to productive ends (Collobert et al., 2011; Mikolov et al., 2013a;b; Peters et al., 2018). N-gram models providing frequencies of pairs, triples, etc. of words in natural text provided further gains on related tasks. However, a very influential paper in 2018, signalling a major shift in the application of machine learning to natural text, advocated for an architecture that has "a more structured memory for handling long-term dependencies in text, compared to alternatives like recurrent networks, resulting in robust transfer performance across diverse tasks." (Radford et al., 2018) This culminated in the application of the Transformer (Vaswani et al., 2017) to the creation of representations through language prediction tasks, motivated by the importance of long-term dependencies in natural language text for not only the choice of *model*, but also *training data*; as Radford et al. note, "Crucially, [BooksCorpus], a common corpus for a multitude of emerging transformer models, contains long stretches of contiguous text, which allows the generative model to learn to condition on long-range information."

Why might 'long stretches of contiguous text', via learning conditioned on that text, lead to success at diverse tasks like natural language inference, question answering, sentence similarity, and classification (Radford et al., 2018, Table 1)? After all, these tasks typically involve very short, independent sections of text.

Solving the Winograd Schema Challenge (WSC) Levesque et al. (2012) seems to require vast amounts of common sense knowledge, and the job of learning long-term dependencies was supposed to help replacing actual knowledge of the world with the proxy knowledge that human-generated text provides. Although language models do well at common sense benchmarks through fine-tuning, when we evaluate them using standard fine-tuning methods with a small, admittedly unreliable 'quick-probe', they do not generalize well to new samples that we offer. On the other hand, a recent zero-shot technique using an idiosyncratic model with several unique architectural and training features shows remarkable consistency and absolute performance on our unreliable quick probe, but also on a family of challenging common sense problems.

## 1.1 Summary of Contributions:

In this paper we investigate the properties of a language model with parameter sharing: `albert-xxlarge-v2`, small in both parameter count and pre-training corpus relative to the field of language models generally. We find that pseudo-log-likelihoods (PLL) and token-normalied PLLs (NormPLL) methods for scoring natural language with this model performs at a mixture of outright state-of-the-art (SOTA) performance and robust, but SOTA performance just for zero-shot methods at a series of recent binary common sense language tasks. The combination of model and method is remarkable consistent, scoring around 75-80% under conditions both designed to be adversarial against language models. The approach is also robust against accidental processes that reduce zero-shot performance in language models generally, such as semantically and syntactically noisy data.

To our knowledge, our results are SOTA for any approach to the TimeDial (Qin et al., 2021) dataset; SOTA for any zero-shot approach to solving the train-xl split of Winogrande (Sakaguchi et al., 2020); SOTA for an average score on the perturbed Winograd set (Abdou et al., 2020); and, SOTA for any zero-shot approach to WSC, with the exception of a reported result in which training and testing sets were mixed. In other cases, our approach is SOTA for zero-shot and competitive with fine-tuned approaches. We provide an explanation for the results and their significance.

## 2 Related Work

### 2.1 Bidirectional vs. unidirectional models

The two most recent GPT papers, 'Language Models are Unsupervised Multitask Learners' (Radford et al., 2019) and 'Language models are few-shot learners' (Brown et al., 2020) identify in their titles the nature or purpose of machine learning models for language with the purposes they put their GPT variants to in their paper. Emphatic titles aside, the most influential fine-tuning papers also advocate for few- and zero-shot results. A more important differentiator between GPT style and NormPLL-suitable models is the significant benefit of a bidirectional masked objective for success with PLL scoring methods over single-directional masked objectives, as Salazar et al. (2020), Zhou et al. (2020), and Ma et al. (2021) show.

### 2.2 The 'quick-probe assumption'

In his discussion of Winograd schemas, Dennett defines what he calls the 'quick-probe assumption': success on a few Winograd schemas in a Turing test-style evaluation ought to indicate generalizability of a computer's ability to make common sense judgements, not merely success at the few examples like it, or examples like it in some superficial way only (Dennett, 1984).

One of us, skeptical of fine-tuning for success at tasks like the Winograd Schema Challenge and similar problems, hand-made a set of 20 sentence pairs[1] prior to collaboration on the present paper.

---

[1] `https://anonymous.4open.science/r/NotSoFineTuning-4620/` `winogradversarial/examples.json` We have reproduced the dataset in its entirety in Appendix A.2.

The purpose of this set of Winograd-style pairs is to test whether fine-tuning can be attacked directly, as follows.

Suppose a training set contains multiple complete pairs, such that reference is shifted every time a sentence has a twin that is different only in some modifier or short phrase. Then perhaps a pair in which reference *isn't* shifted will be scored poorly, if the model is spuriously using the modifier trick. This can be an exploited trick (at least in principle) if, for example, one member of a Winograd schema pair is in the train set, and the other is in the test set [2].

Here is an example from this small, hand-made data set:

1. This is why people are supposed to take salt tablets when $< mask >$ sweat a lot. Answers: people, salt tablets
2. This is why people are supposed to take salt tablets when $< mask >$ sweat a little. Answers: people, salt tablets

By substituting the answers in for the mask above we get two pairs of sentences for a model to score, or assess the relative likelihood of, resulting in two questions of the suitcase/trophy example above. The correct answer above for both examples is 'people', since salt tablets don't sweat.

| Model | Fine-tuned | Zero-Shot |
|---------|-----------|-----------|
| BERT | 45% | 55% |
| RoBERTa | 50% | 60% |
| ALBERT | 55% | **65%** |
| DeBERTa | 50% | 55% |

Table 1: Performance of various transformer models (large versions), Fine-tuning performed on Winogrande.

In Table 1 we compare the performance of a variety of models that have been fine-tuned on the Winogrande, a scaled WSC-variant debiased against RoBERTa (Sakaguchi et al., 2020). We find that the BERT family of language models generally does poorly on this data set when evaluating its fine-tuned discriminator on the data set. On the other hand, using a method of scoring sentences using language models in a manner which is free of hyperparameters, we also score the models – in the second column, there is no training beyond the objective functions of the models during semi-supervised pre-training.

Notice that a single model outperforms the others: `albert-large`. The `albert-xxlarge-v2` variant scores an impressive **80%** on the Winogradversarial dataset we present. It is a well-defined question to ask whether this high value for that last variant is a statistical fluke, or evidence of a robust ability to score binary common sense sentence pairs at a rate of around 80%.

An anonymous reviewer points out that, on 20 coin flips, there is a greater than 25% chance of achieving more than 12 heads, or 60% accuracy. Therefore the results of Table 1 are not particularly meaningful. We agree: these results are not particularly meaningful, by themselves. This paper argues on the basis of new results on very large binary choice and similar data sets that the 80% score achieved by the `albert-xxlarge-v2` is due to its compositionality of representation and corresponding systematicity of behaviour. We also cite independent research that supports our interpretation of our results.

## 2.3 HYPERPARAMETERS AND ZERO-SHOT

A broad survey of machine learning research concludes that demonstrating an ability to innovate in model construction dominates work done in data set collection and cleaning (Sambasivan et al., 2021). Resource constrained researchers are able to use platforms like huggingface to leverage pre-training with novel models contributed by other researchers. PLLs applied to language models for common sense tasks present both opportunities and challenges distinct from this standard approach

---

[2]This is in fact turns out to be the case in the WNLI dataset which is part of the general natural understanding benchmark of SuperGLUE Wang et al. (2019)

to distributed work in NLP. Predictive language model scoring using a pre-trained deep learning model has been used since at least (Linzen et al., 2016), although as we discuss below PLLs seem to display unique benefits for architectures with idiosyncratic features such as parameter sharing and bidirectionality.

Despite its nascent application, scholarly literature has already recognized availability of GPU time for researchers as a limiting factor in applying NormPLLs (our term) for large language models. Laban et al. (2021) explicitly limit their investigation to the smallest, 'base' models of BERT and RoBERTa in their published results. As we demonstrate below, ALBERT requires an order of magnitude more GPU time for NormPLL scoring than BERT, but we nevertheless provide results for a number of important data sets using the 'xxlarge' variant of ALBERT.

In Appendix C we compare the approach we share here to a wide range of common sense tasks, including COPA (Roemmele et al., 2011). The website associated with the COPA dataset contains an ethics injunction for its users with specific imperatives: researchers should not peek at the data set before they evaluating their model on it; and, researchers should only evaluate their method on COPA once.[3] Our zero-shot method scores an impressive 80% on COPA; as we argue below, sensitivity of the method to even extra spaces around punctuation marks necessitates a certain amount of familiarity with the data.

We see critical views of fine-tuning in industry. A recent white paper eschews fine-tuning and even few-shot evaluation for assessing the representational quality of a natural language model because of their potential for spurious results.[4] Research parallelism can produce spurious results simply as a consequence of the number of hypotheses tested.[5] It is beyond the scope of this paper, but there are number of methods, such as Bonferroni correction, that can be used in settings of multiple hypothesis testing. Regardless of one's priors for the compliance of fine-tuning of language models by the research community with statistical best practices, one may find zero-shot measurements of language models more reliable simply because of the fewer points of possible p-hacking, intentional or otherwise.

## 3 METHODS AND RESULTS

### 3.1 METHODS

Here we describe three recent papers that all use some form of PLL or NormPLL, recently published, that do not cite one another – this speaks to a large community of focused and determined researchers in a wide variety of private and public settings. We employ the codebase of the first two approaches in preparation of our own results. For brevity we refer the reader to the papers mentioned below for an explanation of the algorithms.

#### 3.1.1 MLM-SCORING

We first became aware of PLL scoring using language models via Salazar et al. (2020), and to our understanding their arxiv submission of that paper in late 2019 is the first treatment of the approach in the machine learning literature, although we acknowledge that the vast literature is growing ever more quickly. Much of our scoring is performed using the codebase associated with the paper.[6] One key advantage of this codebase is that its use of `mxnet` means that scoring of individual sentences is efficiently shared among multiple GPUs if available. A minor disadvantage is that, in our experience on a managed academic computing platform, is that package compatibility was harder to achieve.

Salazar et al. (2020) style scoring is reported with GPU time for evaluation on an academic computing node with the following characteristics: 32 cores, RAM of 187G or 192000M, 2 x Intel Silver 4216 Cascade Lake @ 2.1GHz, 1 x 480G SSD, and 4 GPUs, all NVIDIA V100 Volta (32G HBM2 memory).

---

[3]See `https://people.ict.usc.edu/~gordon/copa.html`
[4]See https://www.ai21.com/blog/announcing-ai21-studio-and-jurassic-1
[5]For an easy to digest example, see `https://xkcd.com/882/`
[6]See `https://github.com/awslabs/mlm-scoring`

Notably the Salazar et al. (2020) paper treats many topics of interest in machine learning related to language, but does not examine PLL-style scoring on any common sense data sets.

### 3.1.2 CATS SCORING

Zhou et al. (2020) exclusively focuses on the application of NormPLL-style scoring to common sense data sets. What we call the NormPLL algorithm is the Salazar et al. (2020) pseudo-log-likelihood scoring method, but dividing scores by the tokenized length of the expression. We had already completed a number of experiments when finding this codebase, and had already considered the concept of normalization over tokenized length. When comparing Winograd-style pairs, substitutions are usually of similar length – but they may not be.

An advantage of this codebase[7] is that it can be run in any environment that supports the huggingface and pytorch packages. A disadvantage of this approach is that there is no built-in parallelization across multiple GPUs if available; however, because the NormPLL algorithm involves summing over multiple forward passes of language models, it is well-suited to standard MapReduce-style parallelization.

### 3.1.3 ZERO-SHOT WITH HYPERPARAMETERS

Ma et al. (2021) present NormPLLs also under a scoring term (see their $S_{MLM}(T)$ definition) and then augment their performance by providing language models with additional mask-filling training on instances of common sense judgements with tags. It is interesting to note that this results in the presentation of zero-shot results that are qualified with a '95% confidence interval'. Below we compare some of their results with NormPLLs using `albert-xxlarge-v2`, with a fuller picture available in Appendix C.

## 3.2 STATE OF THE ART WITHOUT FINE-TUNING

### 3.2.1 PURELY WINOGRADVERSARIAL DATASETS

We consider the performance of a variety of models on data sets that, with no more than light preprocessing, provide pairs of sentences that are labeled according to super-majority human judgement of common sense, with exactly one right answer per pair.

In a forthcoming paper (anonymous) we demonstrate, using NormPLLs with `albert-xxlarge-v2`, a 6.5% average improvement over the best zero-shot scoring presented in Abdou et al. (2020) over their 'perturbed' Winograd schemas. The perturbed schemas are explicitly designed to reveal the brittleness of language model performance on common sense tasks. We briefly report here pertinent results. $Avg\Delta_{Acc}$. and absolute score for RoBERTa both improved significantly. The $Avg\Delta_{Acc}$. for RoBERTa on perturbed fell from worse than human to better than human; absolute accuracy improved around 4%. ALBERT provided a higher average score than Abdou et al. (2020)'s best reported average score by 11%.

These surprising results prompted further investigation. The code made available with Zhou et al. (2020) makes it a trivial task (given GPU time) to extend their implementation (see their '$Score(S)$' definition) of NormPLLs for the suite of tasks they provide. They test variations and sizes of GPT, BERT, XLNet, and RoBERTa. Table 2 reproduces the best scoring NormPLL and Human metrics from their table along with new results for `albert-xxlarge-v2`.

Note our absolute improvement over their average score by substituting the computationally expensive per parameter ALBERT for RoBERTa. RoBERTa has approximately 50% more parameters and 10 times as much training data as ALBERT. These findings are out of step with the prevailing narrative that bigger is better for both language model size and pre-training corpuses.

Table 3 contains results using PLLs on a variety of language models for the Winograd Schema Challenge data set (Levesque et al., 2012). In these data sets tokenized lengths tend to be similar across sentence pairs, and in these experiments we did not normalize scores when evaluating models. This data set is unusual in that every example contains the name of its author, researchers associated

---

[7]See `https://github.com/XuhuiZhou/CATS`

| | CA | WSC | SM | SMR | SWAG | HellaSwag | ARCT1 | ARCT2 | Average |
|---|---|---|---|---|---|---|---|---|---|
| *roberta-large* | *0.962* | *0.694* | *0.792* | *0.512* | *0.769* | *0.5* | *0.606* | *0.599* | *0.679* |
| albert-xxlarge-v2 | **0.972** | **0.798** | 0.733 | **0.571** | **0.789** | **0.553** | 0.493 | 0.554 | **0.701** |
| *HUMAN* | *0.993* | *0.920* | *0.991* | *0.975* | *0.880* | *0.945* | *0.909* | *0.909* | *0.945* |

Table 2: Comparison of `albert-xxlarge-v2` to best reported model in Zhou et al. (2020). Scored using the Zhou et al. (2020) method.

| Zero-shot model | grad | grande | grad-grande | Model size | GPU time |
|---|---|---|---|---|---|
| xlm-mlm-17-1280 | 55.44 | 52.03 | 3.41 | 1.1GB | 04:36:56 |
| gpt2-345m | 57.19 | 56.40 | 0.79 | 1.4GB | 03:24:50 |
| bert-large-cased-wwm | 65.97 | 57.32 | 8.65 | 1.2GB | 02:55:28 |
| roberta-large | 76.84 | 70.77 | 6.07 | 1.3GB | 03:58:21 |
| albert-xxlarge-v1 | 79.64 | 74.82 | 4.82 | 851MB | 15:30:23 |
| albert-xxlarge-v2 | **81.05** | **76.71** | 4.34 | 851MB | 17:38:25 |

Table 3: PLL zero-shot performance on Winograd (Levesque et al., 2012) and Winogrande (train-xl) (Sakaguchi et al., 2020) data sets for a number of recent large language models. We have sorted by Winograd scores, ascending. Model size in Pytorch `.bin` from https://huggingface.co/models. Scored using the Salazar et al. (2020) library.

with the authors. It also contains results for the train-xl split of the Winogrande data set (Sakaguchi et al., 2020) containing over 44k crowdsourced Winograd schema-style examples.

The train-xl split contains 64,505 sentence comparisons. Each comparison involves scoring two sentences, and the model is scored correct if the higher scored sentence is labeled correct. This results in under 1 seconds of node time per row, or slightly under 0.5 seconds per sentence. This is slow by machine standards, but not slow by human standards. In each row we indicate the difference in score of a given model for the two data sets.

We are not aware of a higher zero-shot score on this Winogrande split. Notice that the value reported in Appendix C for the 'WG' column are reported for the much smaller development set from Winogrande. We are aware of a higher zero-shot score for the Winograd Schema Challenge data set in Brown et al. (2020) – 88.3*–, but that value is asterisked by the authors of that paper because they demonstrated that the web crawled pre-training corpus for GPT-3 is contaminated by portions of the WSC dataset.

### 3.2.2 A RECENT (ALMOST) WINOGRADVERSARIAL DATASET: TIMEDIAL

Each row of the TimeDial dataset (Qin et al., 2021) contains four substitutions into a given sentence, two of which are right and two of which are wrong. The term '2-best accuracy' is defined such that a given row is marked correct iff the scores for the two correct substitutions are both scored higher than the highest scored incorrect substitution. In their paper, the authors describe the best fine-tuned performance for TimeDial on a pre-trained model, achieved with T5 (first row of Table 4), as so low as to question whether fine-tuning is a viable approach to the reasoning over temporal dialogs. Note that our zero-shot approach improves absolute accuracy over their fine-tuning results with about one-third fewer parameters, two orders of magnitude less pre-training data, and no fine-tuning.

Table 4 shows scores for a number of models on the TimeDial data set that includes common sense judgements about the reasonability of judgements about time. Because the text data for these examples are so large, we artificially limit the pool to examples for which both scored passages are less than 450 tokens long once tokenized. This reduces the set by about 5%; in future work, methods like the ones used by Ma et al. (2021) can be used to approximate full NormPLLs for sections of text larger than can be scored on a 32GB GPU; simple windowing is also a solution. Notice the significant increase in run-time for `albert-xxlarge-v2` due to its parameter sharing; at run-time, parameters are 'unrolled' across a larger network than the size on disk would suggest. Using NormPLLs with `albert-xxlarge-v2` produces a score on TimeDial that is, so far as we know, is an absolute SOTA –even when compared to to the best fine-tuned model.

| Model | 2-best Accuracy | Model size | GPU time |
|---|---|---|---|
| *T5-large generation* | *0.748* | 2.75GB | unknown |
| bert-large-cased-whole-word-masking, kws | 0.620 | 1.2GB | 01:23:32 |
| bert-large-cased-whole-word-masking, not kws | 0.619 | 1.2GB | 01:24:23 |
| albert-xxlarge-v2, kws | 0.752 | 851MB | 09:19:02 |
| albert-xxlarge v2, not kws | **0.761** | 851MB | 07:52:56 |

Table 4: PLL zero-shot performance on TimeDial data set (Qin et al., 2021) for a number of recent large language models, with highest fine-tuned score from that paper in italics. Scored using the Salazar et al. (2020) library but with normalization by tokenized length. Dataset filtered to examples with tokenized length less than 450 tokens. Model features are reported from https://huggingface.co/models with model size in Pytorch.bin. 'kws' (short for 'keep weird spaces') indicates that the TimeDial dataset is used as original presented at https://raw.githubusercontent.com/google-research-datasets/TimeDial/main/test.json. 'not kws' indicates the application of a string function to input that removes spaces before punctuation symbols.

### 3.2.3 BRITTLENESS OF THE APPROACH

In Appendix C, Table 5, we provide a full picture of our results comparing zero-shot experiments on CSR benchmarks with large and extra large versions of ALBERT with the best performing model, to our knowledge, reported in the literature corresponding to a RoBERTa-Large model trained on additional synthetic datasets drawn from a combination of knowledge bases including ATOMIC, ConceptNet, WordNet, and Wikidata from Ma et al. (2021). As can be seen, our zero-shot language model approach achieves best results for binary choice tasks, but peforms less well than their approach, which augments language models with in-domain learning. For multiple choice questions with one unique answer among more than two options, our approach is inferior. Some data sets, such as COPA, can be rendered into two candidate sentence form: in this case, our performance is similar to binary choice problems.

Important findings that we highlight are that the while our experimentation demonstrates that without any additional data or knowledge source (which in itself would have invite an opportunity for multiple experimentation, even in the zero-shot regime, i.e., *multitake*) ALBERT pre-trained only on its original pre-training corpora achieves SOTA on a number of the CSR benchmarks (e.g., WSC, Winogrande, HellaSwag), it performs competitively (but sightly worse) on others, and is yet outperformed by a large margin on a few others, the most noticeable of which is on SIQA (-14.89%).

### 3.3 A TALE OF THREE WINOGRADS

Here we draw attention to three quantities that should, abstractly, be identical, but are instead different. The Winograd Schema Challenge is a public dataset that is currently available in `xml` format on the open web.[8] Visiting this site in a modern browser such as Google Chrome results in a nicely formatted series of questions, reproduced in Appendix D. On the other hand, by 'viewing the source' of the rendered `xml`, a different representation can be seen making certain features more obvious of the dataset, also reproduced there.

The second representation makes more clear that there is extra white space in the strings for some fields but not others; in some cases there is extra white space at the front, but not back of a string. Also, there are initial capitalizations in the two answer fields that won't be appropriate when substituted for the pronoun so as to complete scoreable sentences.

Now consider the question: how does the `albert-xxlarge-v2` perform on the data set presented in these figures? Consider table 2: 0.798. In (anonymous), using Abdou et al. (2020)'s presentation, it is 0.796. Finally, according to table 3, it is 0.810. These scores are all supposedly produced using the same method on the same data set. The `roberta-large` scores are, respectively, 0.694, 0.708, and 0.768.

---

[8]See https://cs.nyu.edu/~davise/papers/WinogradSchemas/WSCollection.xml

Here is the source of the discrepancy. The highest scores in both cases, table 3, correspond to PLL scoring on Winograd schema challenge data that we have provided a single python script[9] for downloading from its public location on the Web, and then provides some explicit cleaning and concatenating to produce two individual sentences. The other two scores are produced using pipelines from Zhou et al. (2020) for table 2 and Abdou et al. (2020) for the perturbed results we cite using our method.

Those pipelines included preprocessing of the `xml` into other formats that can be inspected via the repositories for those papers.[10] It is to the credit of the authors of both papers that their pipeline has been made public, including the parts. Thanks to this transparency, we can report problems with both datasets.

The Zhou et al. (2020) Winograd Schema Challenge data for table 2 contains what we call here 'weird spaces'. These are expressions such as `care of john . John` . In addition, it contains numerous odd concatenations, such as `Xenophanesconveys`. Finally, it lower cases some proper names, likely in trying to deal with leading `The`s in answer fields, but not others. The Abdou et al. (2020) data is entirely lower cased. The codebase does not provide the final sentence as it is used for model scoring, but inspecting the `jsonl` reveals many extra spaces before punctuation marks.

## 4 DISCUSSION

### 4.1 RECENT WORK ON COMPOSITIONALITY

Interest in the problem of compositionality has been reinvigorated in the context of the advancing capacities of neural networks for language. Baroni (2020) and Russin et al. (2021) lay out existence proofs, providing clear evidence of the learnability of compositional syntactic and semantic domains. Ontañón et al. (2021) go further, and for a series of synthetic tasks that strongly benefit from compositionality, such as arithmetic, they perform ablation experiments across a number of features of modern Transformer architectures, notably for parameter sharing: an unusual feature of ALBERT.

Ontañón et al. (2021) conclude that weight sharing (sometimes called 'parameter sharing', as in that adopted by the Transformer-based model of ALBERT (Lan et al. (2019)) is, alone, a design decision that "significantly boosts compositional generalization accuracy, and almost all models [with weight sharing] achieve a higher average accuracy across all datasets than their equivalent models [without weight sharing]..."(Ontañón et al., 2021, 6) [11] This use of 'compositionality' has taken over the meaning of 'systematicity', referring to behavioral consistency instead of representational form. Generalization accuracy across binary choice common sense natural language tasks as we have seen across a narrow range for `albert-xxlarge-v2` of about 76%-80% might be partly explained by this result for synthetic data sets.

How might a bidirectional transformer encode generalizable language knowledge? Some recent probes of linguistic generalization measures whether BERT's layers can be interpreted as, to some degree, representing the knowledge expressed by Penn Treebank parse trees (Clark et al., 2019), (Hewitt & Manning, 2019). Another approach offering a metric for assessing parse trees and localizable activation in BERT claims in its title that the model 'Rediscovers the Classical NLP Pipeline' Tenney et al. (2019).

These approaches have self-acknowledged limitations. Clark et al. (2019) and Tenney et al. (2019) both point out the poor performance of BERT on coreference resolution. Hewitt & Manning (2019) highlight that the evidence provided by syntactic probes are neither necessary nor sufficient for finding linguistic representations and their use in downstream behavior. "For this reason, [they] "emphasize the importance of combining structural analysis with behavioral studies...to provide a

---

[9] https://anonymous.4open.science/r/NotSoFineTuning-DB54/Winograd/GetAndCleanWinograd.py

[10] https://github.com/XuhuiZhou/CATS/blob/master/commonsense_ability_test/wsc.txt and https://github.com/mhany90/perturbed-wsc/blob/release/data/dataset/enhanced_wsc_jsons/text_original.jsonl

[11] This paper was released after we had completed the vast majority of our experiments.

more complete picture of what information these models encode and how that information affects performance on downstream tasks." We find their motivating insight reminiscent of Dennett (1991), and endorse the need for behavioral studies that identify generalizable linguistic abilities in language models; structural probes are necessarily incomplete.

Winograd schemas are notable in that many examples involve the production of a sentence that alters parse trees on the basis of a semantic change in a single word. Consider the reference of 'she' for the choice of good/bad in the following (Levesque et al., 2012): *Although they ran at about the same speed, Sue beat Sally because she had such a good/bad start.* These schemas, given robust human performance, suggest that a knowledge of syntax that is separate from world knowledge may not be possible for human language, as suggested by Miller (1999). Winograd schemas belong to a class of coreference resolution problems. Through PLL scoring with `albert-xxlarge-v2`, we have presented a robust body of behavioural evidence that fewer parameters can produce more consistent coreference resolution behaviour than previously recognized.

## 4.2 ALBERT'S UNIQUE ARCHITECTURAL FEATURES

One important consequence of the design decision to incorporate parameter sharing into a transformer architecture is that it trades parameter size for computation time. In other words, parameter sharing yields representations that are space efficient relative to other models, but time *inefficient*. At least some researchers have recently argued, though, that time is a resource that ought to be traded for accuracy benefits (Banino et al., 2021).

In addition to a bidirectional masked language objective, like all BERT descendants, ALBERT also has a sentence order prediction task. This binary categorical objective function is more difficut than the original BERT sentence order prediction task which, as has been widely noted, reduces to topic modeling, for which bag-of-word representations are good approximations. ALBERT's sentence order prediction task corresponds to the problem of determining, for a pair of consecutive sentences, which one comes first in the source text and which one comes second. Consider the following pair of sentences chosen randomly from Wikipedia:

Sentence 1: "Audrey Henshall was born in Oldham, Lancashire in 1927[2] and studied at the University of Edinburgh, graduating with an MA in 1949."

Sentence 2: "From 1960 to 1971 she was the Assistant Keeper of Archaeology at the National Museum of Antiquities of Scotland."

One method to identify that Sentence 2 comes after Sentence 1 and not vice-versa is simply the presence of date information. The English language groups together many different causal relations and human experiences into physical metaphors, as has long been noted in the cognitive science literature (Thelen, 1995). The evidence suggests that ALBERT is an architecture that both excels at the formation of compositional representations, but was also trained with an objective function that encourages learning of asymmetric relations, such as 'before'; furthermore, those relations are implicated across multiple domains of human activity.

## 5 CONCLUSION

The remarkable consistency of ALBERT's performance on Winogradversarial, TimeDial, WSC, and Winogrande datasets is a point of optimism for the generalizability of the performance of language models. A limitation of the current approach is that the robust performance seems to be limited to cases in which a common sense judgement can be expressed as the relative likelihood of two natural language alternatives, a promising avenue for future work.

We emphasize the improvement in both computational efficiency and accuracy that is effected for the TimeDial dataset by cleaning punctuation so as to more closely match normal human conventions. The difference between Grad and Grande (Table 3) across language models measured through PLLs provides early, incomplete evidence for the hypothesis that crowdsourcing common sense data sets produces a measurable decline in data quality. The findings of this paper support the view that attention and care to data is as important as model innovations in machine learning generally, despite academic and industry practice not always matching this ideal (Sambasivan et al., 2021).

## 6 REPRODUCIBILITY

We have prepared a public repository with the files that generated our results tables in .csv form, the .csv scored tables, and scripts to read scores from .csv files.

The repository is available publicly at

https://anonymous.4open.science/r/NotSoFineTuning-4620/

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

# A    APPENDIX A – WINOGRADVERSARIAL: AN ADJECTIVE FOR CRAFT DATASETS

## A.1    WINOGRADVERSARIAL DATA FORMAT

A Winogradversarial instance is defined formally as a tuple $P = \{S, C_1, C_2, K\}$, where $S$ is the sentence with a masked token representing a missing coreferring antecedent, $C_1$ and $C_2$ are the candidates for the correct antecedent for the masked token and $K$ indicates the correct antecedent. Note that both $C_1$ and $C_2$ appear in $S$. $\{S, C_1, C_2\}$ are provided as input for models, which must predict $K$ (e.g., as the output of a binary classification over $C_1, C_2$). Additionally, for every sentence $S$ there is another sentence in the set, $S'$ that differs by $S$ by single word, also known as the *switch word* as in the original Winograd Schema Challenge. (i.e., $S$ has some word $w$, while $S'$ is identical to $S$ except $w$ is replaced by another word, $w'$.) Crucially, while in the WSC, the modifying of $S$ by replacing word $w$ by $w'$ switches the correct label, in Winogradverserial, the correct label remains the same for both $S$ and $S'$. This may turn out to be *adverserial* to models that may be conditioned to switch their prediction decision simply based on the changing of the switch word.

Representative sentences $S$ and $S'$ are the following, respectively, with $w$ and $w'$, underlined:

1. $S =\{$Jordan$\}$ wanted to appear nice to $\{$Jim$\}$ so $\langle$mask$\rangle$ ate some breath mints.
2. $S' =\{$Jordan$\}$ wanted to appear nice to $\{$Jim$\}$ so $\langle$mask$\rangle$ offered some breath mints.

Here, $C_1 = $ Jordan, $C_2 = $ Jim, $K = C_2 = $ Jordan, $w = $ ate, and $w' = $ offered

## A.2    20 QUESTIONS

This is the complete 'winogradversarial' dataset we used to generate the hypothesis for this paper. We have inserted line breaks here in sentences to preserve readability.

```
{"sentence":
    "Jordan wanted to appear nice to Jim so <mask> ate some breath mints",
    "option1": "Jordan",
    "option2": "Jim",
    "answer": "1"}
{"sentence": "Jordan wanted to appear nice to Jim so <mask> offered
    some breath mints",
    "option1": "Jordan",
    "option2": "Jim",
    "answer": "1"}
{"sentence": "I get tons of spam at both locations but gmail incorrectly
    identifies <mask>.",
    "option1": "tons of spam",
    "option2": "both locations",
    "answer": "1"}
{"sentence": "I get tons of spam at both locations but gmail correctly
    identifies <mask>.",
    "option1": "tons of spam",
    "option2": "both locations",
    "answer": "1"}
{"sentence": "Sydney isn't currently better than Jack but <mask> has the
    potential to be.",
    "option1": "Sydney",
    "option2": "Jack",
    "answer": "1"}
```

```
{"sentence": "Sydney isn't currently worse than Jack but <mask> has the
    potential to be.",
    "option1": "Sydney",
    "option2": "Jack",
    "answer": "1"}
{"sentence": "Homes should be prepared for children before you have <mask>.",
    "option1": "homes",
    "option2": "children",
    "answer": "2"}
{"sentence": "Homes should be prepared for children after you have <mask>.",
    "option1": "homes",
    "option2": "children",
    "answer": "2"}
{"sentence": "This is why people are supposed to take salt tablets
    when <mask> sweat a lot.",
    "option1": "people",
    "option2": "salt tablets",
    "answer": "1"}
{"sentence": "This is why people are supposed to take salt tablets
    when <mask> sweat a little.",
    "option1": "people",
    "option2": "salt tablets",
    "answer": "1"}
{"sentence": "Psychologists theorize that people are less comfortable
    when <mask> are given too few choices.",
    "option1": "Psychologists",
    "option2": "people",
    "answer": "2"}
{"sentence": "Psychologists theorize that people are less comfortable
    when <mask> are given too many choices.",
    "option1": "Psychologists",
    "option2": "people",
    "answer": "2"}
{"sentence": "The lemon cake tasted better than the banana muffin because
    <mask> was sweet.",
    "option1": "lemon cake",
    "option2": "banana muffin",
    "answer": "1"}
{"sentence": "The lemon cake tasted better than the banana muffin because
    <mask> was savoury.",
    "option1": "lemon cake",
    "option2": "banana muffin",
    "answer": "1"}
{"sentence": "Mark won the competition over James
    because <mask> was too small.",
    "option1": "Mark",
    "option2": "James",
    "answer": "2"}
{"sentence": "Mark won the competition over James
    because <mask> was too large.",
    "option1": "Mark",
    "option2": "James",
    "answer": "2"}
{"sentence": "I prefer to purchase the purse over the shoe because
    <mask> is too cheap.",
    "option1": "the purse",
    "option2": "the shoe",
    "answer": "2"}
{"sentence": "I prefer to purchase the purse over the shoe because
```

```
    <mask> is too expensive.",
    "option1": "the purse",
    "option2": "the shoe",
    "answer": "2"}
{"sentence": "The TV is more valuable than the Ipad,
    so I decided to sell <mask>.",
    "option1": "the TV",
    "option2": "the Ipad",
    "answer": "1"}
{"sentence": "The TV is more valuable than the Ipad,
    so I decided to buy <mask>.",
    "option1": "the TV",
    "option2": "the Ipad",
    "answer": "1"}
```

## B APPENDIX B – DEBERTA

Below we provide values for various Deberta models on our benchmark suite of Winogradversarial datasets.

Note the warning message below from huggingface when using documented masked language model instantiation of Deberta v1 models. v2 models threw an error when called from the

```
DebertaForMaskedLM
```

method.

```
    Some weights of the model checkpoint at microsoft/deberta-large
    were not used when initializing DebertaForMaskedLM:
    ['deberta.embeddings.position_embeddings.weight', 'config',
    'lm_predictions.lm_head.bias',
    'lm_predictions.lm_head.LayerNorm.weight',
    'lm_predictions.lm_head.dense.weight',
    'lm_predictions.lm_head.LayerNorm.bias',
    'lm_predictions.lm_head.dense.bias']
- This IS expected if you are initializing DebertaForMaskedLM from
the checkpoint of a model trained on another task or with another
architecture (e.g. initializing a BertForSequenceClassification
model from a BertForPreTraining model).
- This IS NOT expected if you are initializing DebertaForMaskedLM
from the checkpoint of a model that you expect to be exactly
identical (initializing a BertForSequenceClassification model
from a BertForSequenceClassification model).
Some weights of DebertaForMaskedLM were not initialized from
the model checkpoint at microsoft/deberta-large and are newly initialized:
['cls.predictions.transform.LayerNorm.bias',
'cls.predictions.transform.LayerNorm.weight',
'cls.predictions.transform.dense.bias',
'cls.predictions.bias',
'cls.predictions.transform.dense.weight',
'cls.predictions.decoder.weight']
You should probably TRAIN this model on a down-stream task to
be able to use it for predictions and inference.
```

## C APPENDIX C – FULL COMPARISON WITH MA ET AL. (2021)

Our preliminary experimentation reveals that, as with the data sets discussed in the main text, the presentation format of the inputs in the datasets themselves results in significant variance of results

(e.g., punctuation details, lower-casing, perspicuity of wording) especially for the models that were not designed with parameter sharing. This latter point suggests two important conclusions; a) the robustness of ALBERT isn't only demonstrateed on syntactic perturbations of a single presentation of benchmark, but it is robust to presentation changes altogether in a benchmark, while other transformer models exhibit more brittleness and b) comparisons betwen ALBERT and such models as in Ma et al. (2021) should be treated as possibly incongruous, as the former may reflect a more universal estimation of the models' performance on common-sense reasoning question answer tasks regardless of presentation or choice of external knowledge source, while the latter the upper-bounded performance of transformer models when under most favourable settings.

| Model | aNLI | CSQA | PIQA | SIQA | WG | COPA | DPR | GAP | PDP | GAP | WinoBias | WGA |
|---|---|---|---|---|---|---|---|---|---|---|---|---|
| bert-large-uncased | 54.96 | 29.32 | 59.19 | 42.12 | 51.14 | 63.8 | 62.1 | 50.19 | 63.75 | 68.68 | 53.75 | |
| roberta-large | 65.01 | 44.71 | 67.9 | 45.24 | 56.35 | 72.4 | 61.0 | 55.78 | 66.25 | 79.48 | 60.0 | |
| xlnet-large-cased | | 44.14 | 61.15 | 45.7 | 55.25 | | | | | 69.88 | | |
| albert-large-v2 | 55.74 | 34.8 | 61.8 | 43.19 | 52.48 | 62.2 | 63.5 | 49.46 | 65.0 | 68.6 | 60.0 | |
| albert-xxlarge-v2 | 66.84 | 52.49 | 70.02 | 48.31 | 62.11 | 79.6 | 78.9 | 58.82 | 81.25 | 81.25 | 65.56 | |
| Ma-GPT2-L (multitake*) | 59.2 | 48.0 | 67.5 | 53.5 | 54.7 | | | | | | | |
| Ma-Roberta-L (multitake*) | 70.5 | 67.4 | 72.4 | 63.2 | 60.9 | | | | | | | |
| Human | 85.6 | 88.9 | 94.9 | 86.9 | 94.1 | | | | | | | |

Table 5: Zero-shot, *single-take* evaluation results of language models that have only been exposed to pre-training corpora via original objective function (i.e., without a selection process based on multiple experimental configurations) across various commonsense tasks. Human performance as well as the best text*multiple-take* model Ma et al. (2021) are included for reference. In the case of Ma et al. (2021), multiple takes corresponded selecting from variants of Roberta-large pre-trained on different Knowledge Graphs, and reporting the best-performing variant in the zero-shot setting.

# D    APPENDIX D – BROWSER RENDERS OF WINOGRAD

1. The city councilmen refused the demonstrators a permit because **they** feared violence.

*Snippet:* **they** feared violence

    A. The city councilmen
    B. The demonstrators

**Correct Answer:** A
**Source:** (Winograd 1972)

Figure 1: An xml entry from the Winograd Schema Challenge rendered in the Chrome browser.

```
7   <schema>
8   <text>
9   <txt1>
10  The city councilmen refused the demonstrators a permit because</txt1>
11  <pron> they </pron>
12  <txt2> feared violence.
13  </txt2>
14  </text>
15  <quote>
16  <quote1></quote1>
17  <pron>they </pron>
18  <quote2>feared violence</quote2>
19  </quote>
20  <answers>
21  <answer>The city councilmen </answer>
22  <answer>The demonstrators </answer>
23  </answers>
24  <correctAnswer> A  </correctAnswer>
25  <source> (Winograd 1972) </source>
26  </schema>
```

Figure 2: An xml entry from the Winograd Schema Challenge viewed through the browser's code editor.

