# OpenReview forum: "An Application of Pseudo-log-likelihoods to Natural Language Scoring"
_ICLR.cc/2022/Conference — ICLR 2022 Submitted_

### Official Review · Reviewer_X2TK · 2021-10-29

**Correctness:** 3
**Technical Novelty And Significance:** 2
**Empirical Novelty And Significance:** 2
**Recommendation:** 5
**Confidence:** 4

**Main Review:**

**Strengths:**

Presents a new adversarial common-sense reasoning dataset which does not conform to the usual “flip a word and change the answer” paired sentence paradigm, and thus may be harder for models trained on such datasets.

Highlights the shortcomings of fine-tuned language models on this adversarial common-sense reasoning dataset, and the relative strength of pretrained models.

Unifies multiple independent research threads on the topic of using pre-trained models for language understanding.

Achieves state-of-the-art performance on the TimeDial dataset using the albert-xxlarge-v2 model.

**Weaknesses:**

Introduction presents a skewed history of the field, lumping together papers focusing on tagging using feedforward networks (Collobert et al., 2011), efficient learning of context-independent word-embeddings (Mikolov et al., 2013), and learning context-dependent word-embeddings using a recurrent architecture (Peters et al., 2018), all under the vague heading of “machine learning techniques applied to unstructured corpora providing frequencies of words”. It then points to the Transformer as heralding a new era in the field, but fails to cite “Attention Is All You Need” by Vaswani et al., which was the seminal paper that introduced the Transformer architecture.

One of the purported main contributions of the paper, the new Winogradversarial dataset, contains only 20 examples, and is therefore unlikely to be widely useful (see below). It purports to address a proposed pitfall of existing WSC datasets wherein the model can “cheat” if it has seen one instance of a paired example in the training set, and another in the test set, that it can know a priori that it should invert its response. However no direct evidence of this being an issue is provided for the WSC or WinoGrande datasets. If it was truly believed to be an issue there are multiple more direct ways of handling it, such as explicitly enforcing train/test disjointness, or even removing one of the two sentences from each pair altogether.

Performance on the Winogradversarial dataset itself, as shown in Table 1, is an unreliable indicator. For a model whose “true” accuracy is p, the number of questions it gets right on the Winogradversarial dataset can be modeled as a binomial random variable X ~ Binom(20, p). Setting p=0.5 (the average performance of the fine-tuned models), this implies that the probability of getting X≥12 (i.e. observing an accuracy of ≥60%) is >25%. This is a substantial fraction of the overall probability mass, and therefore unlikely to be meaningful.

It is difficult to pin down the overall purpose of the paper. Within the two introductory sections, topics discussed include:

1. Strengths/weaknesses of word-frequency based approaches
2. Importance of compositionality
3. Concerns around fine-tuning approaches
4. Concerns around existing WSC datasets
5. The grounding problem
6. The introduction of a new WSC dataset
7. Concerns around current incentives and asymmetries in the field

While I understand that these topics are conceptually overlapping, I would strongly encourage the authors to narrow their focus, so that a more targeted and convincing argument can be made.

As discussed in Section 3.1, the previous work of Salazar et al. (2020), Zhou et al. (2020), and Ma et al. (2021) overlaps heavily with the present work. In fact one of the major contributions of this work, namely showing the performantness of the pretrained albert-xxlarge-v2 on common-sense reasoning datasets, merely falls between the cracks of these other papers. In particular Zhou et al. evaluated common-sense reasoning with a variety of different pretrained Transformer models, just not with AlBERT specifically.

In the discussion section, the decreased performance on WinoGrande compared to Winograd is pointed to as evidence that the crowdsourcing of this dataset led to decreased quality. This is a questionable conclusion given that in the original Sakaguchi et al. (2020) paper introducing WinoGrande, fine-tuning on this dataset prior to evaluating on the WSC data by Levesque et al. yielded dramatic performance improvements, achieving 90.1% performance, only a few percent away from human-level, and much higher than the zero-shot performance numbers provided in this work.

The state-of-the-art performance achieved on the TimeDial dataset is not especially meaningful owing to the dataset’s extreme recency. The dataset has only been publicly available for 3-4 months at the time of writing, and as far as I can tell no fine-tuned model has ever been evaluated on it. This renders the claimed state-of-the-art status less impactful than it would have otherwise been.

Minor Issues:

* Top of Page 2: “We are concerned that **that** the fine-tuning”
* Bottom of Page 2: Overly-verbose footnotes. Don’t need to specify where your example names came from, don’t need to point out that people with brain damage have diminished capacities
* Middle of Page 2: “We take the long-standing response to this argument to be a rejection of premise that only classical logic can implement compositional representations.” - meaning is unclear, consider simpler language
* Repeated: Failure to correctly capitalize “BERT”, “RoBERTa”, “AlBERT”
* Bottom of Page 5: Overly-verbose footnotes. The availability or lack-thereof of academic computing resources is not relevant to the present paper. It is assumed that researchers are following the established principle of not peeking at the evaluation set, this does not need to be explicitly called-out.
* Table 2: Inconsistent presence of zeros before the decimal point
* Table 4: T5 accuracy formatted differently than other models
* Bottom of Page 7: “Because **they** text data for these examples are so large”

**Summary Of The Paper:**

The paper makes multiple independent contributions, including:

1. The addition of a new adversarial common-sense reasoning dataset dubbed “Winogradversarial”
2. The explicit gathering together of multiple research threads all exploring the use of pretrained language models for zero-shot prediction on language-understanding tasks
3. The singling-out of the albert-xxlarge-v2 architecture as performing especially well on zero-shot common-sense reasoning tasks, achieving SOTA on the TimeDial dataset

The overall message of the paper is that the existing framework of fine-tuning models for common-sense reasoning tasks is flawed and prone to overfitting, and that more robust results can be achieved using pretrained models in a zero-shot manner. In particular, doing so with the albert-xxlarge-v2 model achieves state-of-the-art performance on some benchmarks.


**Summary Of The Review:**

While the paper makes inroads in a few different interesting directions (presenting an alternative adversarial WSC-style question format, highlighting advantages of pretrained zero-shot models, achieving SOTA on a recently-released dataset), the connections between the ideas are tenuous, and the individual contributions are not sufficiently impressive to stand on their own. I encourage the authors to pick just one or two of these ideas, and flesh them out in a more thorough, convincing manner. In its present form, I believe this paper should be rejected.

---

> ### Author Response · Authors · 2021-11-16
> **Reply to Reviewer; updated version of paper**
>
> We are grateful to the Reviewer for so carefully reading our paper. As a result of their hard work, we have revised the paper so as to more accurately and clearly communicate its central scientific contributions. We have prepared a revision of the paper that, we hope, addresses some of the reviewer's concerns. Below we explain some of the differences in the latest version of the paper, and again thank the Reviewer for helping make this improved version possible.
>
> We are concerned that our lack of clarity in presenting secondary literature may have impacted the Reviewer's evaluation. The Reviewer writes
>
> > The state-of-the-art performance achieved on the TimeDial dataset is not especially meaningful owing to the dataset’s extreme recency. The dataset has only been publicly available for 3-4 months at the time of writing, and as far as I can tell no fine-tuned model has ever been evaluated on it. This renders the claimed state-of-the-art status less impactful than it would have otherwise been.
>
> We have added two sentences to the paper that summarizes our contribution in the context of the following from Qin et al. (2021):
>
> > Our experiments demonstrate that off the-shelf, pre-trained language models cannot effectively reason about temporal aspects in a dialog, even with domain-specific fine-tuning. Our findings indicate that large-scale pre-trained models even after fine-tuning may not be sufficient for robust temporal reasoning in dialogs, and motivate future research toward modeling temporal concepts over diverse everyday events, and contextual reasoning about them. (Qin et al., 2021, Introduction)
>
> The most recent version of the paper implements almost all of the 'Minor Issues' reported by the Reviewer. In the next few sections we discuss some of the remaining Issues.
>
> On the matter of capitalization, we have endeavoured to achieve internal consistency. We were unable to find a source for the recommended capitalization "AlBERT". Lan et al. (2020) use all-caps, as in "ALBERT". It is to the Reviewer's credit that they scrutinized our writing so carefully. There is the question of whether to use capitalization from scholarly works or codebases when mentioning a huggingface model name, for example; in some cases we have left those lowercase, but are open to discussion about this.
>
> The Reviewer writes:
>
> > The availability or lack-thereof of academic computing resources is not relevant to the present paper.
>
> We have moved the relevant footnote into the main text and made more clear the relevance of academic computing to the present paper. When we refer to Laban et al. (2021), we are thinking of the following. Note that our `NormPLL` is their `MLMS`.
>
> > One key disadvantage of MLMS is its computational cost: scoring requires a forward-pass for each word in the sequence; by contrast, generative
> models usually require a single forward pass. This limits our ability to test large models, and therefore test only base models: BERT-base and RoBERTa base.
>
> The Reviewer writes
>
> > It is assumed that researchers are following the established principle of not peeking at the evaluation set, this does not need to be explicitly called-out.
>
> We agree with the principle that there should always be a presumption of research integrity and have rewritten this passage with reference to risks of fine-tuning that do not contain any such presumption.
>
> We have cited Vaswani et al. (2017) and have tried to clarify that what the works we are citing have in common is in their productive computational applications of unstructured corpora of human generated natural language.
>
> The Reviewer says a number of true and important things about the statistical features of evaluating models on a tiny data set consisting of only 20 members. We have acknowledged those features explicitly in the text, and attempt to more accurately convey the application of the data set in generating interest in more reliable investigation of the class of binary choice common sense tasks we are interested in.
>
> We have made some changes such as removing two paragraphs from the Introduction, and moving one paragraph from the Introduction to the Discussion. We hope this has served to narrow the focus of the paper in the way that the Reviewer intends.
> The Reviewer writes
>
> > one of the major contributions of this work, namely showing the performantness of the pretrained albert-xxlarge-v2 on common-sense reasoning datasets, merely falls between the cracks of these other papers
>
> The ICLR call for papers asks for _applications in... text understanding_. We have refocused our paper to clarify its presentation of a novel application. We have also clarified the significance of our results. Thank you again to the anonymous reviewer for their efforts in improving our paper. Please note the modified title and slightly modified abstract in the new revised version of the paper.

---

> > ### Comment · Reviewer_X2TK · 2021-11-17
> > **Response to Revised Submission**
> >
> > Based on the revised submission I have updated my Recommendation from a 3 to a 5, and updated the Technical Novelty And Significance from a 1 to a 2.
> >
> > The core issue with this paper, which even after the (appreciated) revision is still left unresolved, is that the authors are not contributing a new model, nor are they contributing a novel inference technique, nor are they contributing a large-enough-to-be-generally-useful dataset, nor are they deeply probing the inner working of ALBERT to understand why it performs better. (For examples of such probing, see: [Hewitt & Manning, 2019](https://nlp.stanford.edu/pubs/hewitt2019structural.pdf); [Tenney et al., 2019](https://arxiv.org/abs/1905.05950); [Clark et al., 2019](https://arxiv.org/abs/1906.04341).) They emphasize the SOTA zero-shot results achieved with ALBERT, and assert that this enhanced performance is due to the model's increased capacity for "compositional" reasoning, but provide no direct evidence for this.
> >
> > As it stands, the purported contributions of the revised paper are clearer, however they still do not meet the bar for publication. The paper resides in an unfortunate middle-ground where it achieves SOTA, but not by contributing a new model or inference technique. Even this might be permissible if the authors delved deeply into understanding how/why such an existing model was able to achieve SOTA, but instead their primary insight in this vein (ALBERT uses weight sharing) comes from citing an existing paper, Ontañón et al. (2021). My recommendation would be to use these findings of ALBERT's impressive zero-shot reasoning ability as a jumping-off point for a deeper examination of the model to understand what makes weight-sharing so helpful.
> >
> > Additional Typos:
> > * Middle of page 2: "We find that, when using pseudo-log-likelihoods (PLL) and token- normalied PLLs (NormPLL) methods for scoring natural language with this model, it performs at a mixture of outright state-of-the-art at a series of recent binary common sense language tasks, no- tably hovering at around 75-80% under conditions both designed to be adversarial against language models, but also robust against accidental processes that reduce zero-shot performance in language models generally, such as semantically and syntactically noisy data." - extremely long run-on sentence
> > * Middle of page 3: "whether this high value for *for* that last variant" - duplicate word

---

> > > ### Author Response · Authors · 2021-11-22
> > > **Response to Reviewer; Revised Submission**
> > >
> > > We have uploaded a new version of the paper which corrects the typo and long sentence.
> > >
> > > The Reviewer writes
> > >
> > > > (For examples of such probing, see: Hewitt & Manning, 2019; Tenney et al., 2019; Clark et al., 2019.)
> > >
> > > We thank the reviewer for directing our attention to these papers. We find compelling arguments for the value of our contributions in them, and have added three new paragraphs to Section 4.1 accordingly. We have made room for this discussion by shortening our conclusion section and making additional minor improvements to the paper.

---

> > > > ### Comment · Reviewer_X2TK · 2021-11-22
> > > > **Response to inclusion of BERT-probing resources**
> > > >
> > > > We appreciate the authors' addition of these resources to their paper. While it is nice seeing additional background fleshed out, it does not change this reviewer's core complaint which is that this paper contributes little of substantive novelty, and instead leans primarily on the work of other papers for (1) their model, (2) their inference technique, (3) their datasets (aside from the too-small-to-be-informative Winogradversarial), (4) the analysis identifying weight-sharing as important.
> > > >
> > > > These core issues are not resolved by adding additional material to the Background or Discussion sections, they are resolved by doing additional substantive research. That may mean training a new model, or performing an in-depth quantitive analysis on the pitfalls of existing datasets, or developing a new analytical framework for understanding the benefits of parameter sharing... but whatever it is, it has to be tangible, not just more words.

---

### Official Review · Reviewer_zjM9 · 2021-11-01

**Correctness:** 2
**Technical Novelty And Significance:** 2
**Empirical Novelty And Significance:** 2
**Recommendation:** 3
**Confidence:** 4

**Main Review:**

The paper investigates the performance of the Albert pretrained language model on Winograd-style tasks.
They find that it outperforms other models on some datasets and conclude that this is due to the Albert model reusing parameters.

The analysis of different language models in a zero-shot Winograd setting is an interesting area and the paper reports some strong results for the Albert model.

The paper appears to be more of an opinion piece and lacks a clear novel contribution.
An existing pre-trained language model (Albert) is evaluated on existing datasets (Winograd, Winogrande and Timedial) using well-established methods (PLL, NormPLL).
The finding that Albert outperforms more parameter-rich models like XLM or RoBERTa can be considered unintuitive, but the analysis in the paper is not sufficient to demonstrate the cause of this performance difference.

The paper argues that the better performance of Albert is due to the parameter reuse, which supposedly allows better modeling of language compositionality.
However, evaluation is only performed by comparing to other well-known pre-trained language models. Each of these has different training data, different architecture, different number of parameters and a different training objective. Each of these could be responsible for one model outperforming another on a particular dataset.
In order to draw further conclusions, experiments should be performed by controlling these factors and only changing one component at a time.

Section 2.1 seems to criticise previous papers ("Language Models are Unsupervised Multitask Learners" and "Language models are few-shot learners") for claiming that their titles define the purpose of language models. This criticism seems unnecessary - the papers are only showing that pre-trained language models can be used in such a way, not claiming that they cannot be used in any other way.

The experiments in 2.2, using only 20 sentence pairs, are not adding value to the paper. The dataset is far to small to draw any conclusions based on it.

Section 3.2.1. describes results from a "forthcoming publication". By reproducing the whole table in the appendix, it seems to claim credit for the same contribution in multiple papers. I would recommend anonymously citing the forthcoming paper instead.

Discarding the higher GPT-3 results due to their larger training data does not seem properly motivated. The training data was not controlled for any of the other experiments and it was not ensured that the WSC sentences did not appear in the Albert training set.

The analysis in Section 3.2.3. seems to claim competitive results compared to Ma et al. (2021). However, Table 6 shows that Albert outperforms Ma et al only on 1 out of 5 datasets, which seems to contradict this.

Overall, the paper is rather difficult to follow. The narrative jumps around quite a bit, while important components like datasets, evaluation metrics and baselines are not fully introduced.



**Summary Of The Paper:**

The paper investigates the performance of the Albert pretrained language model on Winograd-style tasks.
They find that it outperforms other models on some datasets and conclude that this is due to the Albert model reusing parameters.

**Summary Of The Review:**

The analysis of different language models in a zero-shot Winograd setting is an interesting area and the paper reports some strong results for the Albert model.

The paper appears to be more of an opinion piece and lacks a clear novel contribution.
An existing pre-trained language model (Albert) is evaluated on existing datasets (Winograd, Winogrande and Timedial) using well-established methods (PLL, NormPLL).
The finding that Albert outperforms more parameter-rich models like XLM or RoBERTa can be considered unintuitive, but the analysis in the paper is not sufficient to demonstrate the cause of this performance difference.

---

> ### Author Response · Authors · 2021-11-16
> **Reply to Reviewer; updated version of paper**
>
> We are grateful to the Reviewer for their comments. Below are responses to some of the more substantive comments.
>
> The Reviewer writes
>
> > The paper appears to be more of an opinion piece and lacks a clear novel contribution.
>
> The paper's contribution is intended to be a novel application that achieves surprising robustness and accuracy across data sets but at the expense of GPU time. We have tried to address this comment and the later comment regarding clarity and presentation of important components with a substantial edit that sharpens the focus of the paper. It is to the Reviewer's credit that they made so many helpful remarks despite our less than optimal framing. The framing of the paper should now be clear as offering a novel application with remarkable robustness and performance across a class of common sense tasks.
>
> On the Reviewer's comments on parameter reuse, we respectfully direct them to the Ontanon et al. (2021) paper that controls for a suite of architectural features in discovering the relative contribution of parameter sharing for the compositionality they observe. We have moved the relevant section to the Discussion in the hopes of highlighting the complementary character of that publication to our application results.
>
> The Reviewer says
>
> > ...claiming that their titles define the purpose of language models. This criticism seems unnecessary - the papers are only showing that pre-trained language models can be used in such a way, not claiming that they cannot be used in any other way.
>
> We have rewritten the related paragraph to merely highlight the relative advantage of bidirectional vs. unidirectional masked language objectives for pseudo log-likelihoods.
>
> The earliest Reviewer had a lengthy discussion of our small data set with only 20 examples. We have tried to make clear that this data set is only of interest for motivating the central question of our paper, and the value of our application.
>
> We have modified the reference to a forthcoming publication and removed the corresponding appendix, as suggested by the Reviewer.
>
> The Reviewer writes
>
> > Discarding the higher GPT-3 results due to their larger training data does not seem properly motivated.
>
> We are thinking of the caption to Table 4.2 of the GPT-3 paper, which reads
>
> > On inspection we find some evidence for contamination of the PIQA and Winograd results, and we mark the corresponding results in Section 3 with an asterisk. We find no evidence that other benchmarks are affected. (Brown et al., 2020)
>
> We have rewritten the relevant sentence to make it more clear that the GPT-3 authors themselves discount their own zero-shot performance on WSC because of the contamination of pre-training data by test data in their case.
>
> The Reviewer writes
>
> > The analysis in Section 3.2.3. seems to claim competitive results compared to Ma et al. (2021). However, Table 6 shows that Albert outperforms Ma et al only on 1 out of 5 datasets, which seems to contradict this.
>
> The section title `Brittleness of the approach` was supposed to flag for the reader that Ma et al. (2021)'s approach is better than ours for the problem types we identify. We have rewritten that section to try and provide greater clarity on this point and are grateful to the Reviewer for pointing this out.
>
> Please note the updated title and abstract in the revised version of the paper.

---

### Official Review · Reviewer_T3k9 · 2021-11-03

**Correctness:** 2
**Technical Novelty And Significance:** 1
**Empirical Novelty And Significance:** 2
**Recommendation:** 3
**Confidence:** 3

**Main Review:**

The main contribution seem to be state-of-the-art zero-shot results on commonsense tasks obtained using albert-xxlarge-v2. However, the authors talk about concepts such as compositionality and I am not sure how this relates the main experiments in the paper.

*Strength*:
- Best zero-shot results on commonsense datasets (as claimed by the authors, I only checked a couple of references to see if that claim holds true)

The main *weakness* of the paper is that the writing is confusing, which makes it difficult to see what the main point of the paper is. The authors seem to argue for multiple statements that show up once in the paper and are then dropped or forgotten. An example from the abstract: "We provide evidence for the following conclusion: a language model with relatively few parameters, trained for relatively few steps, can perform robustly across language tasks in a manner that demonstrates compositionality, at the cost of GPU-time for language evaluation." It is unclear to me where compositionality is being demonstrated in the experiments. (Maybe in the experiments with the 20 sentence pairs? But I wouldn't consider those enough evidence.)

Additionally, some of the statements made by the authors seem wrong or at least misleading:
- The authors argue against finetuning by proving a quote from Bender and Koller (2020) on page 3, but I don't see how the quote is related to finetuning as opposed to general language models.
- The authors state in 3.1.1: "We first became aware of PLL scoring using language models via Salazar et al. (2020), and to our understanding their arxiv submission of that paper in late 2019 is the first treatment of the approach in the machine learning literature" However, I believe this approach has been around much longer as a way to score pairs (or sets) of sentences. For example, Linzen et al. (2016) use effectively the same approach with LSTM language models (https://arxiv.org/pdf/1611.01368.pdf).
I would recommend to revise those statements for future versions of the paper.

**Summary Of The Paper:**

I believe that the main contribution is that the paper shows that albert-xxlarge-v2 is the best zero-shot model on the commonsense datasets (out of multiple models available in huggingface that are being evaluated). However, the authors also seem to argue against finetuning as a general approach for commonsense reasoning, though I don't see how this is backed up by the paper (except for maybe the small experiment with 20 sentence pairs?).

**Summary Of The Review:**

While the authors present new state-of-the-art results, there aren't many insights besides the fact that albert-xxlarge-v2 is a good zero-shot model. The authors could design more experiments backing up their main claims (e.g., about compositionality).

---

> ### Author Response · Authors · 2021-11-16
> **Reply to Reviewer**
>
> We are grateful to the Reviewer for their constructive criticims regarding the framing of our applied results. We have re-framed our empirical findings so as to more clearly highlight the benefits and costs of our approach to using pseudo-log likelihoods with masked language models incorporating a high degree of parameter sharing.
>
> Regarding the perceived centrality of the small, 20 question data set to our paper, we have made it clear that the initial findings are intended only to be an initial probe, no more informative (nor less) than flipping a coin 20 times to see if it is fair.
>
> The quoted sentence from the abstract has been made precise so that it is more obvious to the reader what the most impactful, central result of the paper is. An improvement we have made to the draft addresses the relevance of compositionality to our applied results. We have removed some content and moved the discussion of recent synthetic benchmarks relevant to our applied results to the Discussion section.
>
> The quote from Bender and Koller (2020) in the earlier version of the paper was intended to be understood exactly as the Reviewer describes it: as applying to the practicality of language models in general, fine-tuned or otherwise. We were trying to say that our zero-shot application of pseudo-log likelihoods to language models is a more nuanced alternative to the wholesale rejection of language models as learning common sense. We have removed the reference and the discussion so as to focus the paper.
>
> We have used the freed up space to more carefully trace the secondary literature. The comparison of our best TimeDial results to fine-tuned approaches with much larger models was unclear to other Reviewers.
>
> The Reviewer writes:
>
> > "We first became aware of PLL scoring using language models via Salazar et al. (2020), and to our understanding their arxiv submission of that paper in late 2019 is the first treatment of the approach in the machine learning literature" However, I believe this approach has been around much longer as a way to score pairs (or sets) of sentences. For example, Linzen et al. (2016) use effectively the same approach with LSTM language models (https://arxiv.org/pdf/1611.01368.pdf).
>
> We have examined Linzen et al. (2016), and understand the method there as predicting a feature of a word (singular or plural) due to a preceding initial segment. We take the Reviewer to be referring to the following, in the language modeling section of the paper:
>
> > We evaluate the knowledge that the network has acquired about subject-verb noun agreement using a task similar to the verb inflection task. To perform the task, we compare the probabilities that the model assigns to the two forms of the verb that in fact occurred in the corpus (e.g., write and writes), and select the form with the higher probability.11 As this task is not part of the network’s training objec- tive, and the model needs to allocate considerable resources to predicting each word in the sentence...
>
> We see the similarities with the approach: a language model is applied to scoring binary choice language tasks on the basis of higher probabilities. Seeing the lineage described by the Reviewer, we have added the suggested reference to the current version of the paper. There are a number of differences from the present approach though.
>
> First, as now made clearer in the updated paper, pseudo-log likelihoods can only be poorly approximated by unidirectional language models. Bidirectional language models show a distinct advantage, in a way that has been explained independently by the researchers we cite.
>
> The second difference is the pseudo-log likelihood algorithm itself, which calculates independent model probabilities for every combination of word given context, in the bidirectional masked case. The sum of the logs of those probabilities are then summed to get the pseudo-log likelihood. In the normalized case, we divide by the length of the *tokenized* sentence or passage.
>
> The Reviewer writes
>
> > The authors could design more experiments backing up their main claims (e.g., about compositionality).
>
> We have tried to clarify that independent ablation experiments support the interpretation of our observed robustness through compositionality induced by parameter sharing (Ontanon et al., 2021). Related material has been shortened and moved to the Discussion section.
>
> Please note the modified title and abstract in the revised version of the paper.

---

### Official Review · Reviewer_wa7Y · 2021-11-09

**Correctness:** 1
**Technical Novelty And Significance:** 2
**Empirical Novelty And Significance:** Not applicable
**Recommendation:** 1
**Confidence:** 4

**Main Review:**

Strengths:
The paper seems to introduce a new dataset that could be very useful to the community.

Weaknesses:
The paper is generally poorly written.  Even the main claims are hard to decipher, right from the abstract.  But the entire introduction provides no clear/coherent overview/findings or motivation.  The remainder of the paper continues the same way.  This is a pity, since it seems like the authors have some interesting results to present.  They are too unclear for me to be secure in restating them.
A new dataset is introduced  (winogradversarial), over which the main evidence of the claims of the paper are obtained, without any description of its contents beyond a couple examples and an extremely obscure description of the purpose of its examples.

**Summary Of The Paper:**

The paper demonstrates some evidence that albert-xxlarge-v2's NormPLL has compositional capacity.  The paper does this by introducing the Winogradversarial dataset and documenting performance.

**Summary Of The Review:**

See weaknesses.

---

> ### Author Response · Authors · 2021-11-16
> **Rebuttal revision submitted**
>
> We acknowledge that we could have focused the paper more clearly on its core applied results and have provided an updated paper that, we believe, goes a considerable distance towards correcting the issues the Reviewer describes. Please note the modified title and abstract in the new .PDF.

---

### Author Response · Authors · 2021-12-01
**Table 5 -- correction of two column headers**

In the Appendix, Table 5, there are two column headers that are incorrect.

The second "GAP" should be "Winobias", and the "Winobias" should instead be "Winogender".

In verifying the correct headers we have discovered an error in our representations of the Winogender data. We have included both OCCUPATION/PARTICIPANT and PRONOUN erroneously. For example, the first example is scored as

> The technician told the customer that technician he could pay with cash.

This instead should read

> The technician told the customer that he could pay with cash.

As a result, we expect our Winogender scores are lower than accurate.

---

### Decision · Program_Chairs · 2022-01-20

**Decision:**

Reject

**Comment:**

This paper argues several loosely-related points about the evaluation of pretrained models on commonsense reasoning datasets in the Winograd style, and presents experiments with existing models on several datasets, including a novel 20-example benchmark. All four reviewers struggled to find a clear contribution or theme in this paper that is novel and thorough enough to meet the bar for publication at a selective general-ML venue.

I'd urge the authors to focus in on just one of these points and expand, and to consider submitting to a venue that more narrowly focuses on methods for commonsense reasoning in NLP.